# Post-Operative Infections in Head and Neck Cancer Surgery: Risk Factors for Different Infection Sites

**DOI:** 10.3390/jcm11174969

**Published:** 2022-08-24

**Authors:** Giancarlo Pecorari, Giuseppe Riva, Andrea Albera, Ester Cravero, Elisabetta Fassone, Andrea Canale, Roberto Albera

**Affiliations:** Division of Otorhinolaryngology, Department of Surgical Sciences, University of Turin, 10126 Turin, Italy

**Keywords:** head and neck surgery, head and neck cancer, post-operative infection, surgical site infection, pneumonia, bacteremia, urinary tract infection

## Abstract

Background: Post-operative infections in head and neck cancer (HNC) surgery represent a major problem and are associated with an important increase in mortality, morbidity, and burden on the healthcare system. The aim of this retrospective observational study was to evaluate post-operative infections in HNC surgery and to analyze risk factors, with a specific focus on different sites of infection. Methods: Clinical data about 488 HNC patients who underwent surgery were recorded. Univariate and multivariate analyses were performed to identify risk factors for post-operative infections. Results: Post-operative infections were observed in 22.7% of cases. Respiratory and surgical site infections were the most common. Multiple site infections were observed in 3.9% of cases. Considering all infection sites, advanced stage, tracheotomy, and higher duration of surgery were risk factors at multivariate analysis. Median hospital stay was significantly longer in patients who had post-operative infection (38 vs. 9 days). Conclusions: Post-operative infections may negatively affect surgical outcomes. A correct identification of risk factors may help the physicians to prevent post-operative infections in HNC surgery.

## 1. Introduction

Healthcare-associated infections (HAIs) are reported to be associated with an important increase in mortality, morbidity, and burden on the healthcare system, especially in the post-operative setting. In the clinical practice, the most frequent types of HAIs are: surgical site infection (SSI), bloodstream infection, respiratory infection, and urinary tract infection [1].

Post-operative HAIs in head and neck cancer (HNC) surgery represent a critical problem and may affect oncological results. Indeed, they can determine a delay in adjuvant radiotherapy (RT) and/or chemotherapy (CT) [2]. Despite the use of prophylactic antibiotics and the best pre- and post-operative care, HAIs incidence is still high due to the expanding microbial resistance [3].

SSIs following head and neck oncological surgery are the main post-operative HAIs and range from 10 to 50% [2,3,4,5,6,7,8,9,10,11,12,13,14,15,16,17,18,19,20,21]. Respiratory infection after HNC surgery ranges between 7 and 40%, urinary tract infection between 2.1 and 6.1%, and bacteremia from 0.7 to 13.8% [2,22,23,24,25,26,27]. HAIs are associated with prolonged hospitalization, readmission rates, poor cosmetic results and mortality [2,3,18,22,24,27]. Consequently, direct costs rise [25].

Risk factors for SSIs have been analyzed in a number of studies and included higher American Society of Anesthesiologists (ASA) score, comorbidities, smoking, alcohol consumption, previous RT or CT, advanced tumor stage, longer duration of surgery, blood loss and/or anemia, hypoalbuminemia and/or malnutrition (weight loss), presence of tracheotomy, flap reconstruction, and clean-contaminated surgery [4,6,10,11,12,14,15,16,17,18,19,20,21,28]. Although different authors analyzed these risk factors, significant disagreements with heterogeneous results exist in the literature.

Only a few studies investigated risk factors for respiratory infections and identified age, chronic obstructive pulmonary disease (COPD), tumor stage, smoking, weight loss as significant factors for infections occurrence [2,22,23,24,27]. Only two studies evaluated risk factors for infections from different sites considered together (all HAIs) [2,22]. In particular, previous radiotherapy, anemia, salvage surgery, tracheotomy, longer surgery duration, microvascular reoperation < 72 h, and flap loss were identified as risks factors for HAIs [2,22]. Finally, to the best of our knowledge, risk factors for multiple site infections have never been investigated.

The aim of this retrospective observational study was to evaluate post-operative infections in HNC surgery and to analyze risk factors identifying the patients at higher risk, with a specific focus on different sites of infection.

## 2. Materials and Methods

The study sample included 488 patients who underwent HNC surgery at our Division between January 2015 and May 2022. Inclusion criteria were: age > 18 years, and surgery for cancer of oral cavity, pharynx, larynx, paranasal sinuses, salivary glands. Minor procedures, such as biopsies, were excluded from the study. The study was conducted in accordance with the Declaration of Helsinki and approved by the Institutional Review Board (protocol code 0021433, date of approval 26 February 2021). All the patients were contacted and provided written informed consent.

We collected the following clinical data: age, sex, tumor site, smoking, alcohol consumption, body mass index (BMI), comorbidities (allergies, diabetes mellitus, COPD, chronic kidney disease—CKD, cirrhosis), chronic systemic corticosteroid and/or immunosuppressive therapy, tumor node metastasis (TNM) stage, previous RT and/or CT for head and neck cancer, American Society of Anesthesiologists (ASA) score, type of surgical procedure (clean or clean-contaminated surgery), flap reconstruction, tracheotomy, duration of surgery, presence of peripherally inserted central catheter (PICC) or other central venous catheter (CVC), presence of nasogastric feeding tube, post-operative Intensive Care Unit (ICU) stay, pre- and postoperative hemoglobin (Hb), hospitalization length, and site of the infection (surgical site, respiratory system, urinary system, bloodstream, other sites). Antibiotic prophylaxis was performed in clean-contaminated surgery. Intravenous amoxicillin with clavulanic acid 2.2 g was usually used during surgical procedure and every 8 h for at least 3 days after surgery. Intravenous clarithromycin 500 mg was administered to patients who were allergic to amoxicillin.

HAIs are defined according to the United States Centers for Disease Control and Prevention (CDC) guidelines, until the 30th post-operative day [1].

All statistical analyses were carried out using the Statistical Package for Social Sciences, version 20.0 (IBM Corporation, Armonk, NY, USA). The Kolmogorov–Smirnov test demonstrated a non-Gaussian distribution of variables, so non-parametric tests were used. A descriptive analysis of all data was performed, and they were reported as medians and interquartile range (IQR), or percentages. The Mann–Whitney U test was used to assess differences between groups in the mean of continuous variables, while the Chi-squared test was used for categorical variables. Logistic regression (forced entry method) was used for multivariate analysis. If less than 10 cases per each variable were present, multivariate analysis was not performed because of insufficient statistical power. A *p* < 0.05 was considered statistically significant.

## 3. Results

Median age was 66 years (IQR 15 years). Fifty-one percent of patients was older than 65 years. Median BMI was 24.44 (IQR 5.31). Table 1 and Table 2 report patient and tumor characteristics. In particular, 226 (46.3%) patients had an ASA score of 1 or 2, while 262 (53.7%) cases had an ASA score of 3 or 4. Tumors were diagnosed in early stage (I–II) in 53.2% of cases, whereas 46.8% of patients had an advanced tumor (stage III–IV).

Surgical treatments are highlighted in Table 3. Eighty-one (16.6%) patients went to ICU after surgery. ICU stay ranged between 1 and 3 days. Surgery was classified as clean in 83 (17.0%) cases and clean-contaminated in 405 (83%) cases. PICC/CVC was used in 190 (38.9%) patients, while nasogastric feeding tube was positioned in 269 (55.1%) cases.

Median duration of surgery was 195 min (IQR 210 min), and median hospital stay was 13 days (IQR 21 days). Median pre-operative Hb was 14.2 g/dL (IQR 2.0 g/dL), while it was 12.0 g/dL (IQR 2.8 g/dL) at first post-operative day.

Post-operative infections were observed in 111 (22.7%) patients, after a median time of 10 days (IQR 10.25 days). Respiratory infections were the most common (Figure 1). Moreover, multiple site infections were observed in 19 (3.9%) cases, with bacteremia and respiratory infection being the most frequent association. Other sites of infection included gastrointestinal tract (three cases) and male genital system (one case).

Significant risk factors for post-operative infection (all the sites) at univariate analyses were higher ASA score, advanced stage, longer duration of surgery, tracheotomy, clean-contaminated surgical procedure, flap reconstruction, ICU stay, lower post-operative Hb (first day after surgery), nasogastric feeding tube, and presence of PICC/CVC (*p* < 0.05). Median Hb at the days of infection diagnosis was 10.2 g/dL (IQR 2.4 g/dL), lower than levels at first post-operative day (1.8 g/dL lower on average). Table 4 reports the percentages of infections for each site and categorical risk factor, while Table 5 shows statistical significance for risk factors according to infection sites. Older patients had a higher risk of SSIs.

Multivariate analysis (logistic regression) was performed only considering all the infections, because there were too few cases for each infection site to perform statistical analysis. Since the presence of nasogastric feeding tube was significantly associated with the type of surgery (quite all the patients who underwent major clean-contaminated surgery had a feeding tube), it was removed from multivariate analysis. Significant risk factors for post-operative infection were advanced stage, tracheotomy, and longer duration of surgery (*p* < 0.05, Table 6).

Infections were observed in 10.9% and 36.0% of cases in patients with early (stage I–II) and advanced (stage III–IV) cancer, respectively (*p* = 0.048). Patients with tracheotomy had infection in 39.3% of cases, compared to 6.5% in those without tracheotomy (*p* = 0.039). Median duration of surgery was 320 and 150 min in patient with and without post-operative infection, respectively (*p* = 0.001). Median hospital stay was significantly longer in patients who had post-operative infection (38 vs. 9 days, *p* < 0.001, Figure 2).

## 4. Discussion

Post-operative infections represent an important issue in HNC surgery, leading to a longer hospital stay, a delay of adjuvant treatments, worse survival rates, and higher costs [2,25]. The increase of microbial resistance is worsening HAIs, reducing the efficacy of antibiotic prophylaxis. For example, methicillin-resistant Staphylococcus aureus (MRSA) has been emerging as a major cause of SSIs in HNC patients [3].

SSIs following HNC surgery are the main post-operative HAIs and range from 10 to 50% [2,3,4,5,6,7,8,9,10,11,12,13,14,15,16,17,18,19,20,21]. Respiratory infection, bacteremia, and urinary tract infection were observed in 7–40%, 2.1–6.1%, and 0.7–13.8% of cases, respectively [2,22,23,24,25,26,27]. Our study showed HAIs rates in agreement with the literature and located near lower range values. In particular, we observed SSIs in 9.2% of cases, respiratory infections in 12.5%, urinary tract infections in 1.4%, and bacteremia in 3.7%. Globally, we observed post-operative HAIs in 22.7% of patients. The heterogeneous results present in literature are due to different selection of patients in the studies. Indeed, some papers included only patients who underwent flap reconstruction, with a specific focus on free flaps. Since several studies identified flap reconstruction as a HAI risk factor, papers that included only patients who underwent reconstructive surgery reported a higher rate of post-operative infections [2,6,14].

Risk factors have been investigated for SSIs in several studies and for respiratory infections in some papers. However, there is no accordance among studies to identify the same risk factors. Concerning SSIs, the most reported risk factors were ASA score, comorbidities (e.g., diabetes mellitus), smoking, alcohol consumption, previous RT or CT, tumor stage, duration of surgery, blood loss and/or anemia, hypoalbuminemia and/or malnutrition (weight loss), tracheotomy, flap reconstruction, and clean-contaminated surgery [4,6,10,11,12,14,15,16,17,18,19,20,21,28]. Therefore, some variables related to the patient and others related to surgical procedure negatively impact on post-operative HAIs. In particular, malnutrition incidence in HNC patients ranges from 30% to 50% and together with surgical stress, lead to immunosuppression that results in a higher risk of infectious complications and a decrease in survival rates [19].

Our univariate analysis regrading all the infections showed that higher ASA score, tumor site (oral cavity and larynx/hypopharynx), advanced stage, presence of PICC/CVC or nasogastric feeding tube, clean-contaminated surgery, tracheotomy, flap reconstruction, higher duration of surgery, ICU stay, and lower post-operative Hb were potential risk factors. Multivariate analysis confirmed advanced stage, tracheotomy, and higher duration of surgery as risk factors for post-operative infections. The last two parameters were also identified as risk factors by Ramos-zayas et al. in a sample of 65 patients who underwent free flap reconstruction [2]. On the other hand, Tjoa et al. found that age > 65 years and clean-contaminated surgery were risk factors in a large sample of 715 patients who had flap reconstruction [22]. We did not find statistical significance for some clinical parameters, such as comorbidities, immunosuppression and previous RT and/or CT. Our data suggest that HNC surgery may be safe also for patients who underwent organ transplantation and were immunosuppressed. However, no definitive conclusion can be drawn because of small numbers.

Concerning infections of specific sites (surgical site, respiratory system, urinary tract, bloodstream), univariate analyses highlighted slightly different potential risk factors compare to all the infections. Our results about SSIs were in agreement with the literature. The main difference compared to all the infections was the lack of significance for ASA score, tumor site and ICU stay.

Respiratory infections had the same potential risk factors compared to SSIs, adding ICU stay. This is in agreement with the literature. Indeed, a recent meta-analysis showed a significant increase in the post-operative respiratory infections and sepsis in patients admitted to ICU compared with non-ICU setting [29].

Bacteremia and multiple site infections had similar risk factors. In particular, previous CT became significant, while post-operative anemia was not. Our study demonstrated that multiple site infections were more frequent in patients with higher ASA score, advanced tumor stage, previous CT, presence of PICC/CVC or nasogastric feeding tube, clean-contaminated surgery, tracheotomy, flap reconstruction, higher duration of surgery, and ICU stay.

According to literature, we found a longer hospitalization in patents with HAIs. In particular, median hospital stay was 38 days in patients who had post-operative infection, while it was 9 days in non-infected patients. As reported by Penel et al., post-operative infections with longer hospital stay lead to higher direct costs [25].

The strength of this study is the analysis of post-operative HAIs from different sites and the evaluation of multiple site infections. Indeed, at our knowledge, this is the first study that investigated risk factors for each site, including multiple infections. The main limit is the number of cases not sufficient for multivariate analysis for every infection site.

Further studies on large samples are mandatory to obtain reliable results from multivariate analyses. The exact identification of risk factors may help the physicians to prevent post-operative HAIs in HNC surgery.

## 5. Conclusions

Surgery represents one of the main treatments for HNC. However, post-operative infections may negatively affect surgical outcomes. Respiratory and surgical site infections are the most frequent infectious complications. Advanced stage, tracheotomy, and higher duration of surgery are risk factors considering all infection sites. Slight differences have been observed for specific anatomical sites. Future studies are necessary to identify risk factors exactly and thus help surgeons to reduce post-operative infections. In particular, higher attention should be paid to patients with greater risk, who may benefit from longer or different antibiotic prophylaxis.

## Figures and Tables

**Figure 1 jcm-11-04969-f001:**
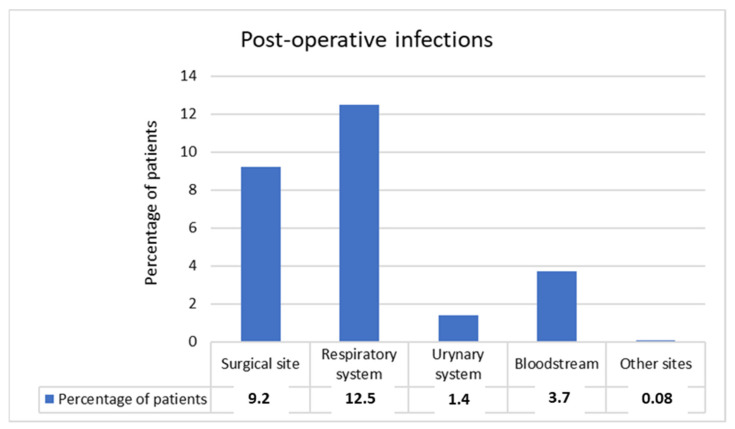
Sites of post-operative infections (percentages on 488 patients).

**Figure 2 jcm-11-04969-f002:**
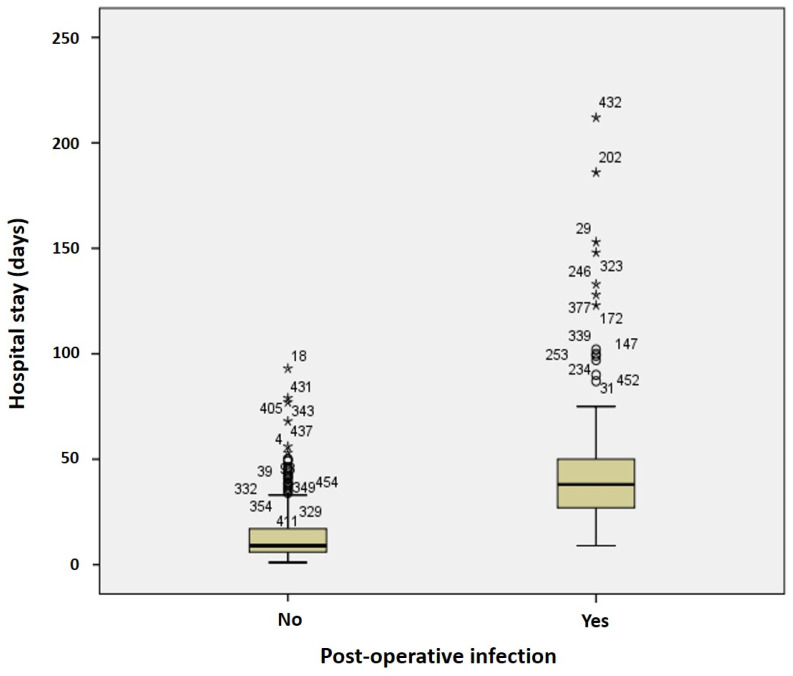
Median hospital stay according to infection status. Outliers are present in the figure as asterisks and circles.

**Table 1 jcm-11-04969-t001:** Patients’ characteristics.

Patient and Tumor Characteristics	N° (%)
** *Sex* **	
Male	381 (78.1)
Female	107 (21.9)
** *BMI* **	
Low (<18.5)	24 (4.9)
Normal (18.5–25)	246 (50.4)
High (>25)	218 (44.7)
** *Smoking* **	
Never	104 (21.3)
Former	211 (43.2)
Active	173 (35.5)
** *Alcohol consumption ** **	243 (49.8)
** *Allergies *** **	99 (20.3)
** *COPD* **	64 (13.1)
** *Cirrhosis* **	13 (2.7)
** *CKD* **	18 (3.7)
** *Diabetes mellitus* **	64 (13.1)
** *Previous transplantation **** **	6 (1.2)
** *Chronic systemic corticosteroid therapy ***** **	16 (3.3)
** *Immunosuppression* **	3 (0.6)
** *Previous radiation therapy* **	118 (24.2)
** *Previous chemotherapy* **	71 (14.5)
** *ASA score* **	
1	44 (9.0)
2	187 (38.3)
3	235 (48.2)
4	22 (4.5)

* Refers to current alcohol consumption (>2 drinks per day for men and >1 drink per day for women). ** Refers to any type of allergy (environmental, drugs, foods) proven by prick tests and/or serological exams. *** Three patients had solid organ transplantation and underwent immunosuppression, while three patients had stem cell transplantation. **** For allergic diseases (13 cases; prednisone 5–10 mg per day for more than 30 days/year) or previous solid organ transplantation (3 cases; prednisone 25 mg per day). BMI, body mass index; COPD, chronic obstructive pulmonary disease; CKD, chronic kidney disease; ASA, American Society of Anesthesiologists.

**Table 2 jcm-11-04969-t002:** Tumor characteristics.

Patient and Tumor Characteristics	N° (%)
** *Tumor site* **	
Nasal cavities and paranasal sinuses	36 (7.4)
Nasopharynx	1 (0.2)
Oral cavity	145 (29.6)
Oropharynx	57 (11.7)
Larynx	177 (36.3)
Hypopharynx	23 (4.7)
Salivary glands	37 (7.6)
Unknown primary	12 (2.5)
** *T* **	
0	28 (5.7)
1	199 (40.8)
2	102 (20.9)
3	93 (19.1)
4	66 (13.5)
** *N* **	
0	344 (70.5)
1	44 (9.0)
2	76 (15.6)
3	24 (4.9)
** *Stage* **	
I	190 (38.9)
II	70 (14.3)
III	94 (19.3)
IV	134 (27.5)

T, tumor; N, node.

**Table 3 jcm-11-04969-t003:** Surgical procedures.

Surgery	N° (%)
Ethmoido-maxillectomy	6 (12.3)
Partial maxillectomy	25 (51.2)
Subtotal/total maxillectomy	10 (2.0)
Partial glossectomy	43 (8.8)
Hemiglossectomy	27 (5.5)
Subtotal/total glossectomy	8 (1.6)
Glossectomy with mandibulectomy	10 (2.0)
Oral floor cancer removal	16 (3.3)
Cheek mucosa cancer removal	10 (2.0)
Lip cancer removal	11 (2.3)
Retromolar trigone cancer removal	9 (1.8)
Partial pharyngectomy	43 (8.8)
Cordectomy	65 (13.3)
Partial laryngectomy	41 (8.4)
Total laryngectomy/pharyngolaryngectomy	82 (16.8)
Parotidectomy	31 (6.4)
Submandibular gland cancer removal	2 (0.4)
Neck dissection	263 (53.9)
Reconstruction with pedicled flap	70 (14.3)
Reconstruction with free flap	30 (6.1)
Tracheotomy	242 (49.6)

**Table 4 jcm-11-04969-t004:** Percentage of post-operative infection for each risk factor in different sites.

Risk Factors	All Infections	Surgical Site Infections	Respiratory Infections	Bloodstream	Multiple Site Infection
Age > 65 years (yes/no)	23.7/21.8	7.2/11.3	14.4/10.5	4.0/3.3	3.6/4.2
Sex (male/female)	23.8/18.7	9.7/7.5	12.9/11.2	3.9/2.8	4.2/2.8
Smoking (active/former or never)	25.4/21.3	11.5/7.9	12.1/12.7	4.6/3.2	4.0/3.8
Alcohol consumption	25.9/19.6	10.7/7.7	14.4/10.6	3.3/4.1	4.9/2.8
BMI (high/low or normal)	22.0/23.3	6.9/11.1	12.3/12.6	3.2/4.1	3.2/4.4
Allergies (yes/no)	28.3/21.3	11.1/8.7	15.1/11.8	5.0/3.3	3.0/4.1
COPD (yes/no)	21.9/22.9	10.9/8.9	12.5/12.5	1.6/4.0	4.7/3.8
Cirrhosis (yes/no)	30.8/22.5	7.7/9.3	15.4/12.4	7.7/3.6	0.0/4.0
CKD (yes/no)	11.1/23.2	5.5/9.4	0.0/12.9	5.5/3.6	0.0/4.0
Diabetes mellitus (yes/no)	26.6/22.2	9.4/9.2	18.7/11.5	3.1/3.8	6.2/3.5
Previous transplantation (yes/no)	33.3/22.6	0.0/9.3	0.0/12.6	16.7/3.5	0.0/3.9
Chronic corticosteroid therapy (yes/no)	37.5/22.2	6.2/9.3	18.7/12.3	6.2/3.6	0.0/4.0
Immunosuppression (yes/no)	33.3/22.7	0.0/9.3	0.0/12.6	0.0/3.7	0.0/3.9
Previous radiation therapy (yes/no)	17.8/24.3	7.6/9.7	10.2/13.2	5.1/3.2	5.9/3.2
Previous chemotherapy (yes/no)	26.8/22.1	8.4/9.3	16.9/11.7	8.4/2.9	8.4/3.1
ASA score (III–IV/I–II)	26.3/18.6	9.5/8.8	14.5/10.2	5.3/1.8	5.3/2.2
Stage (advanced/early)	36.1/10.8	13.5/5.4	20.4/5.4	5.6/1.9	6.1/1.9
PICC/CVC (yes/no)	36.8/13.7	13.7/6.4	21.0/7.0	8.4/1.0	7.9/2.0
Nasogastric feeding tube (yes/no)	35.3/7.3	15.2/1.8	18.6/5.0	5.6/1.4	6.3/0.9
Surgical procedure (clean-cont./clean)	26.7/3.6	11.1/0.0	14.6/2.4	4.4/0.0	4.7/0.0
Tracheotomy (yes/no)	39.2/6.5	14.5/4.1	22.7/2.4	7.0/0.4	7.4/0.4
Flap reconstruction (yes/no)	42.7/17.9	18.7/6.9	26.0/9.2	5.2/3.3	8.3/2.8
ICU stay (yes/no)	45.7/18.2	14.8/8.1	25.9/9.8	9.9/2.4	9.9/2.7

BMI, body mass index; COPD, chronic obstructive pulmonary disease; CKD, chronic kidney disease; ASA, American Society of Anesthesiologists; PICC, peripherally inserted central catheter; CVC, central venous catheter; ICU, intensive care unit.

**Table 5 jcm-11-04969-t005:** Risk factors for different sites of post-operative infections (*p* values at univariate analyses).

Risk Factors	All Infections	Surgical Site Infections	Respiratory Infections	Bloodstream	Multiple Site Infection
Age	0.839	** *0.015 ** **	0.147	0.364	0.468
Sex	0.258	0.480	0.649	0.583	0.510
Smoking (active)	0.294	0.186	0.858	0.416	0.897
Alcohol consumption	0.095	0.261	0.205	0.644	0.235
BMI	0.565	0.140	0.685	0.807	0.865
Allergies	0.141	0.467	0.372	0.421	0.619
COPD	0.858	0.611	1.000	0.333	0.725
Cirrhosis	0.484	0.847	0.750	0.438	0.462
CKD	0.230	0.584	0.102	0.668	0.384
Diabetes mellitus	0.435	0.964	0.105	0.797	0.296
Previous transplantation	0.534	0.432	0.352	0.090	0.620
Chronic corticosteroid therapy	0.152	0.676	0.442	0.580	0.413
Immunosuppression	0.661	0.580	0.511	0.734	0.727
Previous radiation therapy	0.141	0.492	0.379	0.355	0.189
Previous chemotherapy	0.383	0.808	0.225	** *0.021 ** **	** *0.032 ** **
ASA score	** *0.014 ** **	0.782	0.215	** *0.001 ** **	** *0.050 ** **
Tumor site	** *0.008 ** **	0.458	0.536	0.985	0.893
Stage	** *<0.001 ** **	** *<0.001 ** **	** *<0.001 ** **	0.105	** *0.003 ** **
PICC/CVC	** *<0.001 ** **	** *0.007 ** **	** *<0.001 ** **	** *<0.001 ** **	** *<0.001 ** **
Nasogastric feeding tube	** *<0.001 ** **	** *<0.001 ** **	** *<0.001 ** **	** *0.014 ** **	** *0.002 ** **
Type of surgical procedure	** *<0.001 ** **	** *0.001 ** **	** *0.002 ** **	** *0.050 ** **	** *0.044 ** **
Tracheotomy	** *<0.001 ** **	** *<0.001 ** **	** *<0.001 ** **	** *<0.001 ** **	** *<0.001 ** **
Flap reconstruction	** *<0.001 ** **	** *<0.001 ** **	** *<0.001 ** **	0.381	** *0.012 ** **
Duration of surgery	** *<0.001 ** **	** *<0.001 ** **	** *<0.001 ** **	** *<0.001 ** **	** *<0.001 ** **
ICU stay	** *<0.001 ** **	0.057	** *<0.001 ** **	** *0.001 ** **	** *0.002 ** **
Pre-operative Hb	0.440	0.855	0.255	0.801	0.993
Post-operative Hb (1st day)	** *<0.001 ** **	** *0.006 ** **	** *<0.001 ** **	0.186	0.182

* *p* < 0.05 (Mann–Whitney U test for continuous variables and Chi-squared test for categorical variables). BMI, body mass index; COPD, chronic obstructive pulmonary disease; CKD, chronic kidney disease; ASA, American Society of Anesthesiologists; PICC, peripherally inserted central catheter; CVC, central venous catheter; ICU, intensive care unit; Hb, hemoglobin.

**Table 6 jcm-11-04969-t006:** Multivariate analysis regarding risk factors for post-operative infection (all sites).

Risk Factor	*p*-Value
ASA score	0.227
Tumor site	0.652
Stage	** *0.048 ** **
PICC/CVC	0.219
Type of surgical procedure	0.203
Tracheotomy	** *0.039 ** **
Flap reconstruction	0.965
Duration of surgery	** *0.001 ** **
ICU stay	0.402
Post-operative Hb (1st day)	0.224

* *p* < 0.05 (logistic regression). ASA, American Society of Anesthesiologists; PICC, peripherally inserted central catheter; CVC, central venous catheter; ICU, intensive care unit; Hb, hemoglobin.

## Data Availability

The data presented in this study are available on request from the corresponding author.

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
