# Peer review of "Post-Operative Infections in Head and Neck Cancer Surgery: Risk Factors for Different Infection Sites"

_jcm, 2022, doi:10.3390/jcm11174969_

Round 1

Reviewer 1 Report

The authors have considered a number of HAIs in this large study and provide useful information.  However, the manuscript can be improved if the authors address the issues listed below.

1.  Methods: Authors should give duration of follow up for cases, i.e. 30 days or to hospital discharge.  HAI definitions are cited as being those from the CDC, but CDC definitions are for 30 day postoperative follow up.  If follow up is only until hospital discharge, that should be noted.

2.  Methods:  The authors state that "ethical review and approval were waived for this study", but that is very unusual since the authors accessed protected health information (PHI). The authors should provide the journal with a statement from their institutional review board (IRB) explaining that the IRB was aware that the authors had access to PHI yet waived the need for IRB approval of this study.  

3.  There is no table showing the actual number of patients who developed an HAI: Table 4 only gives the P values for univariate analysis.  The actual numbers of each type of infection (SSI, Respiratory, Bloodstream, UTI, other, multiple sites) for each of the risk factors in Table 4 should be provided. 

4.  The types of infections (and number of patients with each type) should be provided for the "other" category of HAIs.  According to Figure 1, this category included "0.8%" of the 488 patients (39 patients).

5. Table 1 shows "chronic corticosteroid therapy" and "immunosuppression" as different categories.  In Methods or in footnote to table, please explain what was included in "immunosuppression" and give figures for numbers of patients on chronic corticosteroid therapy who were receiving immunosuppressive doses of these corticosteroids.  There were 6 patients who had received "previous transplantations", yet only 3 patients were listed in the "immunosuppression" row.  Explain, since most patients who have received organ transplantations remain on immunosuppressive therapy.   Also explain the number of patients in the "previous transplantation" group had solid organ transplantation versus stem cell transplantation. 

6.  Table 1 lists "Alcohol consumption" but doesn't further define.  Please explain what is meant by this (current, number of drinks per week, etc).  

7.  Table 1 lists "Allergies" in approximately 20% of patients.  Do the authors mean environmental allergies?  What factor in the past medical history did the authors use as inclusion criteria for "allergies"?  Note this in Methods or footnote to the table.

8.  Table 1 should list numbers of patients with low BMI and high BMI, since BMI is listed as a variable in Table 4.

9.  Although there were only 6 patients with prior organ transplanation and 3 with "immunosuppression", the authors conclude (Discussion) that "Our data suggest that HNC surgery is safe also for patients who underwent organ transplantation and were immunosuppressed."  How can the authors draw any such conclusion when so few patients are in these categories? This statement should be changed to explain that no conclusion can be drawn when numbers are so small. 

10.  Table 4 lists "smoking" but which group is included here, based on categories listed in Table 1 ("never, former, active")?

Author Response

Dear Editor,

We thank you for the opportunity to revise our paper. Hereby you will find attached a point-by-point answer to each reviewers’ suggestion. Changes were highlighted.

  • Reviewer 1
  • The authors have considered a number of HAIs in this large study and provide useful information.
  • Thanks for your positive comments and useful suggestions.

  • Methods: Authors should give duration of follow up for cases, i.e. 30 days or to hospital discharge. HAI definitions are cited as being those from the CDC, but CDC definitions are for 30 day postoperative follow up.  If follow up is only until hospital discharge, that should be noted.
  • We specified that follow-up was until 30th post-operative day.

  • Methods: The authors state that "ethical review and approval were waived for this study", but that is very unusual since the authors accessed protected health information (PHI). The authors should provide the journal with a statement from their institutional review board (IRB) explaining that the IRB was aware that the authors had access to PHI yet waived the need for IRB approval of this study.
  • We are sorry for the error. The approval by IRB was obtained and protocol number was reported.

  • There is no table showing the actual number of patients who developed an HAI: Table 4 only gives the P values for univariate analysis. The actual numbers of each type of infection (SSI, Respiratory, Bloodstream, UTI, other, multiple sites) for each of the risk factors in Table 4 should be provided.
  • The percentage of each type of infection for each risk factor was provided (new table 4).

  • The types of infections (and number of patients with each type) should be provided for the "other" category of HAIs. According to Figure 1, this category included "0.8%" of the 488 patients (39 patients).
  • We added the types of infections and their number for the “other” category of HAIs. Sorry for the error in Figure 1, that was corrected (0.08%, 4 cases).

  • Table 1 shows "chronic corticosteroid therapy" and "immunosuppression" as different categories. In Methods or in footnote to table, please explain what was included in "immunosuppression" and give figures for numbers of patients on chronic corticosteroid therapy who were receiving immunosuppressive doses of these corticosteroids.  There were 6 patients who had received "previous transplantations", yet only 3 patients were listed in the "immunosuppression" row.  Explain, since most patients who have received organ transplantations remain on immunosuppressive therapy.   Also explain the number of patients in the "previous transplantation" group had solid organ transplantation versus stem cell transplantation.
  • We added more specification in the text and as footnotes of table 1. Three patients had solid organ transplantation and underwent immunosuppression, while 3 patients had stem cell transplantation.

  • Table 1 lists "Alcohol consumption" but doesn't further define. Please explain what is meant by this (current, number of drinks per week, etc).
  • The definition was added as a footnote to table 1.

  • Table 1 lists "Allergies" in approximately 20% of patients. Do the authors mean environmental allergies?  What factor in the past medical history did the authors use as inclusion criteria for "allergies"?  Note this in Methods or footnote to the table.
  • We specified that it refers to any type of allergy proven by prick tests and/or serological exams (footnote to table 1).

  • Table 1 should list numbers of patients with low BMI and high BMI, since BMI is listed as a variable in Table 4.
  • Number of patients with low and high BMI was added in table 1.

  • Although there were only 6 patients with prior organ transplanation and 3 with "immunosuppression", the authors conclude (Discussion) that "Our data suggest that HNC surgery is safe also for patients who underwent organ transplantation and were immunosuppressed." How can the authors draw any such conclusion when so few patients are in these categories? This statement should be changed to explain that no conclusion can be drawn when numbers are so small.
  • The statement was changed according to the suggestion.

  • Table 4 lists "smoking" but which group is included here, based on categories listed in Table 1 ("never, former, active")?
  • We specified that active smoking was considered in table 4.

We remain at your disposal for any further clarification.

Best regards,

The Authors

Reviewer 2 Report

The authors present an interesting retrospective study on post-operative infections in head and neck cancer. The topic has been analysed in literature, but they present a wide cohort and contemporary evaluate different conditions and characteristics. The background is complete and includes all relevant information. The methods are well written and solid. The results are interesting, and the discussion is well written, with an interesting comparison with literature. I would only ask the authors, if possible, to complete the text with a more incisive conclusion: have this study changed or could change the approach to HNC patients?

Author Response

Dear Editor,

We thank you for the opportunity to revise our paper. Hereby you will find attached a point-by-point answer to each reviewers’ suggestion. Changes were highlighted.

  • Reviewer 2
  • The authors present an interesting retrospective study on post-operative infections in head and neck cancer. The topic has been analysed in literature, but they present a wide cohort and contemporary evaluate different conditions and characteristics. The background is complete and includes all relevant information. The methods are well written and solid. The results are interesting, and the discussion is well written, with an interesting comparison with literature. I would only ask the authors, if possible, to complete the text with a more incisive conclusion: have this study changed or could change the approach to HNC patients?
  • Thanks for your positive comments and useful suggestion. We included a more incisive conclusion: higher attention should be paid to patients with greater risk of infection, who may benefit from longer or different antibiotic prophylaxis.

We remain at your disposal for any further clarification.

Best regards,

The Authors

Reviewer 3 Report

Comments for the Authors:

1.       The introduction needs to be elaborated to get a better understanding of SSI (Surgical site Infections) and HAIs (Healthcare-associated Infections). The current risk factors involved in HAIs can be explained in more depth.

2.       The statistics applied in the tables should be mentioned below table as footnotes. The abbreviations used in tables need an explanation as well.

3.  Table no 4 and 5 need improvement as the table content is not clearly depicted. For instance, age is a risk factor for post-operative infections (POIs) but the highest susceptible age group for POIs is not mentioned. In similar ways, sex is a risk factor for POIs but the gender more susceptible to infections is not indicated either in the text or table.

4.       The table title should be clearly explanatory. However, it is not reflecting the purpose of the table clearly.

5.       The references need some corrections. Please correct reference no 2 (page no missing), reference no 7 (Authors' names should not be written in all uppercase) and reference no 13 (page no missing). 

Author Response

Dear Editor,

We thank you for the opportunity to revise our paper. Hereby you will find attached a point-by-point answer to each reviewers’ suggestion. Changes were highlighted.

  • Reviewer 3
  • The introduction needs to be elaborated to get a better understanding of SSI (Surgical site Infections) and HAIs (Healthcare-associated Infections). The current risk factors involved in HAIs can be explained in more depth.
  • Thanks for your useful suggestions. We explained risk factors in more depth in the Introduction section.

  • The statistics applied in the tables should be mentioned below table as footnotes. The abbreviations used in tables need an explanation as well.
  • The statistics and the abbreviations were added as table footnotes.

  • Table no 4 and 5 need improvement as the table content is not clearly depicted. For instance, age is a risk factor for post-operative infections (POIs) but the highest susceptible age group for POIs is not mentioned. In similar ways, sex is a risk factor for POIs but the gender more susceptible to infections is not indicated either in the text or table.
  • A better description was added in the text to explain results. Moreover, a new table 4 with percentages of infections for each site and risk factor was added.

  • The table title should be clearly explanatory. However, it is not reflecting the purpose of the table clearly.
  • The table titles were changed to be clearly explanatory.

  • The references need some corrections. Please correct reference no 2 (page no missing), reference no 7 (Authors' names should not be written in all uppercase) and reference no 13 (page no missing).
  • The references were corrected.

We remain at your disposal for any further clarification.

Best regards,

The Authors